# Momentum-Based Variance Reduction in Non-Convex SGD

**Ashok Cutkosky**
Google Research
Mountain View, CA, USA
ashok@cutkosky.com

**Francesco Orabona**
Boston University
Boston, MA, USA
francesco@orabona.com

## Abstract

Variance reduction has emerged in recent years as a strong competitor to stochastic gradient descent in non-convex problems, providing the first algorithms to improve upon the converge rate of stochastic gradient descent for finding first-order critical points. However, variance reduction techniques typically require carefully tuned learning rates and willingness to use excessively large "mega-batches" in order to achieve their improved results. We present a new algorithm, STORM, that does not require any batches and makes use of adaptive learning rates, enabling simpler implementation and less hyperparameter tuning. Our technique for removing the batches uses a variant of momentum to achieve variance reduction in non-convex optimization. On smooth losses $F$, STORM finds a point $\boldsymbol{x}$ with $\mathbb{E}[\|\nabla F(\boldsymbol{x})\|] \leq O(1/\sqrt{T} + \sigma^{1/3}/T^{1/3})$ in $T$ iterations with $\sigma^2$ variance in the gradients, matching the optimal rate and without requiring knowledge of $\sigma$.

## 1   Introduction

This paper addresses the classic stochastic optimization problem, in which we are given a function $F : \mathbb{R}^d \to \mathbb{R}$, and wish to find $\boldsymbol{x} \in \mathbb{R}^d$ such that $F(\boldsymbol{x})$ is as small as possible. Unfortunately, our access to $F$ is limited to a stochastic function oracle: we can obtain sample functions $f(\cdot, \xi)$ where $\xi$ represents some sample variable (e.g. a minibatch index) such that $\mathbb{E}[f(\cdot, \xi)] = F(\cdot)$. Stochastic optimization problems are found throughout machine learning. For example, in supervised learning, $\boldsymbol{x}$ represents the parameters of a model (say the weights of a neural network), $\xi$ represents an example, $f(\boldsymbol{x}, \xi)$ represents the loss on an example, and $F$ represents the training loss of the model.

We do not assume convexity, so in general the problem of finding a true minimum of $F$ may be NP-hard. Hence, we relax the problem to finding a critical point of $F$ – that is a point such that $\nabla F(\boldsymbol{x}) = 0$. Also, we assume access only to stochastic gradients evaluated on arbitrary points, rather than Hessians or other information. In this setting, the standard algorithm is stochastic gradient descent (SGD). SGD produces a sequence of iterates $\boldsymbol{x}_1, \ldots, \boldsymbol{x}_T$ using the recursion

$$\boldsymbol{x}_{t+1} = \boldsymbol{x}_t - \eta_t \boldsymbol{g}_t, \tag{1}$$

where $\boldsymbol{g}_t = \nabla f(\boldsymbol{x}_t, \xi_t)$, $f(\cdot, \xi_1), \ldots, f(\cdot, \xi_T)$ are i.i.d. samples from a distribution $D$, and $\eta_1, \ldots \eta_T \in \mathbb{R}$ are a sequence of learning rates that must be carefully tuned to ensure good performance. Assuming the $\eta_t$ are selected properly, SGD guarantees that a randomly selected iterate $\boldsymbol{x}_t$ satisfies $\mathbb{E}[\|\nabla F(\boldsymbol{x}_t)\|] \leq O(1/T^{1/4})$ [9].

Recently, *variance reduction* has emerged as an improved technique for finding critical points in non-convex optimization problems. Stochastic variance-reduced gradient (SVRG) algorithms also produce iterates $x_1, \ldots, x_T$ according to the update formula (1), but now $\boldsymbol{g}_t$ is a *variance reduced* estimate of $\nabla F(\boldsymbol{x}_t)$. Over the last few years, SVRG algorithms have improved the convergence rate to critical points of non-convex SGD from $O(1/T^{1/4})$ to $O(1/T^{3/10})$ [2, 21] to $O(1/T^{1/3})$ [8, 31]. Despite

this improvement, SVRG has not seen as much success in practice in non-convex machine learning problems [5]. Many reasons may contribute to this phenomenon, but two potential issues we address here are SVRG's use of *non-adaptive learning rates* and reliance on *giant batch sizes* to construct variance reduced gradients through the use of low-noise gradients calculated at a "checkpoint". In particular, for non-convex losses SVRG analyses typically involve carefully selecting learning rates, the number of samples to construct the gradient on the checkpoint points, and the frequency of update of the checkpoint points. The optimal settings balance various unknown problem parameters exactly in order to obtain improved performance, making it especially important, and especially difficult, to tune them.

In this paper, we address both of these issues. We present a new algorithm called STOchastic Recursive Momentum (STORM) that achieves variance reduction through the use of a variant of the momentum term, similar to the popular RMSProp or Adam momentum heuristics [24, 13]. Hence, our algorithm does not require a gigantic batch to compute checkpoint gradients – in fact, our algorithm does not require any batches at all because it *never needs to compute a checkpoint gradient*. STORM achieves the *optimal convergence rate* of $O(1/T^{1/3})$ [3], and it uses an *adaptive learning rate* schedule that will automatically adjust to the variance values of $\nabla f(\boldsymbol{x}_t, \xi_t)$. Overall, we consider our algorithm a significant qualitative departure from the usual paradigm for variance reduction, and we hope our analysis may provide insight into the value of momentum in non-convex optimization.

The rest of the paper is organized as follows. The next section discusses the related work on variance reduction and adaptive learning rates in non-convex SGD. Section 3 formally introduces our notation and assumptions. We present our basic update rule and its connection to SGD with momentum in Section 4, and our algorithm in Section 5. Finally, we present some empirical results in Section 6 and concludes with a discussion in Section 7.

## 2  Related Work

Variance-reduction methods were proposed independently by three groups at the same conference: Johnson and Zhang [12], Zhang et al. [30], Mahdavi et al. [17], and Wang et al. [27]. The first application of variance-reduction method to non-convex SGD is due to Allen-Zhu and Hazan [2]. Using variance reduction methods, Fang et al. [8], Zhou et al. [31] have obtained much better convergence rates for critical points in non-convex SGD. These methods are very different from our approach because they require the calculation of gradients at checkpoints. In fact, in order to compute the variance reduced gradient estimates $\boldsymbol{g}_t$, the algorithm must periodically stop producing iterates $\boldsymbol{x}_t$ and instead generate a very large "mega-batch" of samples $\xi_1, \ldots, \xi_N$ which is used to compute a checkpoint gradient $\frac{1}{N} \sum_{i=1}^{N} \nabla f(\boldsymbol{v}, \xi_i)$ for an appropriate checkpoint point $\boldsymbol{v}$. Depending on the algorithm, $N$ may be as large as $O(T)$, and typically no smaller than $O(T^{2/3})$. The only exceptions we are aware of are SARAH [18, 19] and iSARAH [20]. However, their guarantees do not improve over the ones of plain SGD, and they still require at least one checkpoint gradient. Independently and simultaneously with this work, [25] have proposed a new algorithm that does improve over SGD to match our same convergence rate, although it does still require one checkpoint gradient. Interestingly, their update formula is very similar to ours, although the analysis is rather different. We are not aware of prior works for non-convex optimization with reduced variance methods that completely avoid using giant batches.

On the other hand, *adaptive learning-rate* schemes, that choose the values $\eta_t$ in some data-dependent way so as to reduce the need for tuning the values of $\eta_t$ manually, have been introduced by Duchi et al. [7] and popularized by the heuristic methods like RMSProp and Adam [24, 13]. In the non-convex setting, adaptive learning rates can be shown to improve the convergence rate of SGD to $O(1/\sqrt{T} + (\sigma^2/T)^{1/4})$, where $\sigma^2$ is a bound on the variance of $\nabla f(\boldsymbol{x}_t)$ [16, 28, 22]. Hence, these adaptive algorithms obtain much better convergence guarantees when the problem is "easy", and have become extremely popular in practice. In contrast, the only variance-reduced algorithm we are aware of that uses adaptive learning rates is [4], but their techniques apply only to convex losses.

## 3  Notation and Assumptions

In the following, we will write vectors with bold letters and we will denote the inner product between vectors $\boldsymbol{a}$ and $\boldsymbol{b}$ by $\boldsymbol{a} \cdot \boldsymbol{b}$.

Throughout the paper we will make the following assumptions. We assume access to a stream of independent random variables $\xi_1, \ldots, \xi_T \in \Xi$ and a function $f$ such that for all $t$ and for all $x$, $\mathbb{E}[f(\boldsymbol{x}, \xi_t)|\boldsymbol{x}] = F(\boldsymbol{x})$. Note that we access two gradients on the same $\xi_t$ on two different points in each update, like in standard variance-reduced methods. In practice, $\xi_t$ may denote an i.i.d. training example, or an index into a training set while $f(\boldsymbol{x}, \xi_t)$ indicates the loss on the training example using the model parameter $\boldsymbol{x}$. We assume there is some $\sigma^2$ that upper bounds the noise on gradients: $\mathbb{E}[\|\nabla f(\boldsymbol{x}, \xi_t) - \nabla F(\boldsymbol{x})\|^2] \leq \sigma^2$.

We define $F^\star = \inf_{\boldsymbol{x}} F(\boldsymbol{x})$ and we will assume that $F^\star > -\infty$. We will also need some assumptions on the functions $f(\boldsymbol{x}, \xi_t)$. Define a differentiable function $f : \mathbb{R}^d \to \mathbb{R}$ to be $G$-Lipschitz iff $\|\nabla f(\boldsymbol{x})\| \leq G$ for all $x$, and $f$ to be $L$-smooth iff $\|\nabla f(\boldsymbol{x}) - \nabla f(\boldsymbol{y})\| \leq L\|\boldsymbol{x} - \boldsymbol{y}\|$ for all $x$ and $y$. We assume that $f(\boldsymbol{x}, \xi_t)$ is differentiable, and $L$-smooth as a function of $\boldsymbol{x}$ with probability 1. We will also assume that $f(\boldsymbol{x}, \xi_t)$ is $G$-Lipschitz for our adaptive analysis. We show in appendix B that this assumption can be lifted at the expense of adaptivity to $\sigma$.

## 4    Momentum and Variance Reduction

Before describing our algorithm in details, we briefly explore the connection between SGD with momentum and variance reduction.

The stochastic gradient descent with momentum algorithm is typically implemented as

$$\boldsymbol{d}_t = (1-a)\boldsymbol{d}_{t-1} + a\nabla f(\boldsymbol{x}_t, \xi_t)$$
$$\boldsymbol{x}_{t+1} = \boldsymbol{x}_t - \eta\boldsymbol{d}_t,$$

where $a$ is small, i.e. $a = 0.1$. In words, instead of using the current gradient $\nabla F(\boldsymbol{x}_t)$ in the update of $\boldsymbol{x}_t$, we use an exponential average of the past observed gradients.

While SGD with momentum and its variants have been successfully used in many machine learning applications [13], it is well known that the presence of noise in the stochastic gradients can nullify the theoretical gain of the momentum term [e.g. 29]. As a result, it is unclear how and why using momentum can be better than plain SGD. Although recent works have proved that a variant of SGD with momentum improves the non-dominant terms in the convergence rate on convex stochastic least square problems [6, 11], it is still unclear if the actual convergence rate can be improved.

Here, we take a different route. Instead of showing that momentum in SGD works in the same way as in the noiseless case, i.e. giving accelerated rates, we show that *a variant of momentum can provably reduce the variance of the gradients*. In its simplest form, the variant we propose is:

$$\boldsymbol{d}_t = (1-a)\boldsymbol{d}_{t-1} + a\nabla f(\boldsymbol{x}_t, \xi_t) + (1-a)(\nabla f(\boldsymbol{x}_t, \xi_t) - \nabla f(\boldsymbol{x}_{t-1}, \xi_t)) \qquad (2)$$
$$\boldsymbol{x}_{t+1} = \boldsymbol{x}_t - \eta\boldsymbol{d}_t \ . \qquad\qquad\qquad (3)$$

The only difference is the that we add the term $(1-a)(\nabla f(\boldsymbol{x}_t, \xi_t) - \nabla f(\boldsymbol{x}_{t-1}, \xi_t))$ to the update. As in standard variance-reduced methods, we use two gradients in each step. However, we do not need to use the gradient calculated at any checkpoint points. Note that if $\boldsymbol{x}_t \approx \boldsymbol{x}_{t-1}$, then our update becomes approximately the momentum one. These two terms will be similar as long as the algorithm is actually converging to some point, and so we can expect the algorithm to behave exactly like the classic momentum SGD towards the end of the optimization process.

To understand why the above updates delivers a variance reduction, consider the "error in $\boldsymbol{d}_t$" which we denote as $\boldsymbol{\epsilon}_t$:
$$\boldsymbol{\epsilon}_t := \boldsymbol{d}_t - \nabla F(\boldsymbol{x}_t) \ .$$

This term measures the error we incur by using $\boldsymbol{d}_t$ as update direction instead of the correct but unknown direction, $\nabla F(\boldsymbol{x}_t)$. The equivalent term in SGD would be $\mathbb{E}[\|\nabla f(\boldsymbol{x}_t, \xi_t) - \nabla F(\boldsymbol{x}_t)\|^2] \leq \sigma^2$. So, if $\mathbb{E}[\|\boldsymbol{\epsilon}_t\|^2]$ decreases over time, we have realized a variance reduction effect. Our technical result that we use to show this decrease is provided in Lemma 2, but let us take a moment here to appreciate why this should be expected intuitively. Considering the update written in (2), we can obtain a recursive expression for $\boldsymbol{\epsilon}_t$ by subtracting $\nabla F(\boldsymbol{x}_t)$ from both sides:

$$\boldsymbol{\epsilon}_t = (1-a)\boldsymbol{\epsilon}_{t-1} + a(\nabla f(\boldsymbol{x}_t, \xi_t) - \nabla F(\boldsymbol{x}_t))$$
$$+ (1-a)(\nabla f(\boldsymbol{x}_t, \xi_t) - \nabla f(\boldsymbol{x}_{t-1}, \xi_t) - (\nabla F(\boldsymbol{x}_t) - \nabla F(\boldsymbol{x}_{t-1}))) \ .$$

---
**Algorithm 1** STORM: STOchastic Recursive Momentum
---
1: **Input:** Parameters $k$, $w$, $c$, initial point $\boldsymbol{x}_1$
2: Sample $\xi_1$
3: $G_1 \leftarrow \|\nabla f(\boldsymbol{x}_1, \xi_1)\|$
4: $\boldsymbol{d}_1 \leftarrow \nabla f(\boldsymbol{x}_1, \xi_1)$
5: $\eta_0 \leftarrow \frac{k}{w^{1/3}}$
6: **for** $t = 1$ **to** $T$ **do**
7: $\quad$ $\eta_t \leftarrow \frac{k}{(w + \sum_{i=1}^t G_t^2)^{1/3}}$
8: $\quad$ $\boldsymbol{x}_{t+1} \leftarrow \boldsymbol{x}_t - \eta_t \boldsymbol{d}_t$
9: $\quad$ $a_{t+1} \leftarrow c\eta_t^2$
10: $\quad$ Sample $\xi_{t+1}$
11: $\quad$ $G_{t+1} \leftarrow \|\nabla f(\boldsymbol{x}_{t+1}, \xi_{t+1})\|$
12: $\quad$ $\boldsymbol{d}_{t+1} \leftarrow \nabla f(\boldsymbol{x}_{t+1}, \xi_{t+1}) + (1 - a_{t+1})(\boldsymbol{d}_t - \nabla f(\boldsymbol{x}_t, \xi_{t+1}))$
13: **end for**
14: Choose $\hat{\boldsymbol{x}}$ uniformly at random from $\boldsymbol{x}_1, \ldots, \boldsymbol{x}_T$. (In practice, set $\hat{\boldsymbol{x}} = \boldsymbol{x}_T$).
15: **return** $\hat{\boldsymbol{x}}$
---

Now, notice that there is good reason to expect the second and third terms of the RHS above to be small: we can control $a(\nabla f(\boldsymbol{x}_t, \xi_t) - \nabla F(\boldsymbol{x}_t))$ simply by choosing small enough values $a$, and from smoothness we expect $(\nabla f(\boldsymbol{x}_t, \xi_t) - \nabla f(\boldsymbol{x}_{t-1}, \xi_t) - (\nabla F(\boldsymbol{x}_t) - \nabla F(\boldsymbol{x}_{t-1})))$ to be of the order of $O(\|\boldsymbol{x}_t - \boldsymbol{x}_{t-1}\|) = O(\eta \boldsymbol{d}_{t-1})$. Therefore, by choosing small enough $\eta$ and $a$, we obtain $\|\boldsymbol{\epsilon}_t\| = (1-a)\|\boldsymbol{\epsilon}_{t-1}\| + Z$ where $Z$ is some small value. Thus, intuitively $\|\boldsymbol{\epsilon}_t\|$ will decrease until it reaches $Z/a$. This highlights a trade-off in setting $\eta$ and $a$ in order to decrease the numerator of $Z/a$ while keeping the denominator sufficiently large. Our central challenge is showing that it is possible to achieve a favorable trade-off in which $Z/a$ is very small, resulting in small error $\boldsymbol{\epsilon}_t$.

## 5  STORM: STOchastic Recursive Momentum

We now describe our stochastic optimization algorithm, which we call STOchastic Recursive Momentum (STORM). The pseudocode is in Algorithm 1. As described in the previous section, its basic update is of the form of (2) and (3). However, in order to achieve adaptivity to the noise in the gradients, both the stepsize and the momentum term will depend on the past gradients, à la AdaGrad [7].

The convergence guarantee of STORM is presented in Theorem 1 below.

**Theorem 1.** *Under the assumptions in Section 3, for any $b > 0$, we write $k = \frac{bG^{\frac{2}{3}}}{L}$. Set $c = 28L^2 + G^2/(7Lk^3) = L^2(28 + 1/(7b^3))$ and $w = \max\left((4Lk)^3, 2G^2, \left(\frac{ck}{4L}\right)^3\right) = G^2 \max\left((4b)^3, 2, (28b + \frac{1}{7b^2})^3/64\right)$. Then, STORM satisfies*

$$\mathbb{E}[\|\nabla F(\hat{\boldsymbol{x}})\|] = \mathbb{E}\left[\frac{1}{T}\sum_{t=1}^T \|\nabla F(\boldsymbol{x}_t)\|\right] \leq \frac{w^{1/6}\sqrt{2M} + 2M^{3/4}}{\sqrt{T}} + \frac{2\sigma^{1/3}}{T^{1/3}},$$

*where $M = \frac{8}{k}(F(\boldsymbol{x}_1) - F^\star) + \frac{w^{1/3}\sigma^2}{4L^2k^2} + \frac{k^2c^2}{2L^2}\ln(T+2)$.*

In words, Theorem 1 guarantees that STORM will make the norm of the gradients converge to 0 at a rate of $O(\frac{\ln T}{\sqrt{T}})$ if there is no noise, and in expectation at a rate of $\frac{2\sigma^{1/3}}{T^{1/3}}$ in the stochastic case. We remark that we achieve both rates automatically, without the need to know the noise level nor the need to tune stepsizes. Note that the rate when $\sigma \neq 0$ matches the optimal rate [3], which was previously only obtained by SVRG-based algorithms that require a "mega-batch" [8, 31].

The dependence on $G$ in this bound deserves some discussion - at first blush it appears that if $G \to 0$, the bound will go to infinity because the denominator in $M$ goes to zero. Fortunately, this is not so: the resolution is to observe that $F(\boldsymbol{x}_1) - F^\star = O(G)$ and $\sigma = O(G)$, so that the numerators of $M$ actually go to zero at least as fast as the denominator. The dependence on $L$ may be similarly non-intuitive: as $L \to 0$, $M \to \infty$. In this case this is actually to be expected: if $L = 0$, then there

are no critical points (because the gradients are all the same!) and so we cannot actually find one. In general, $M$ should be regarded as an $O(\log(T))$ term where the constant indicates some inherent hardness level in the problem.

Finally, note that here we assumed that each $f(x, \xi)$ is $G$-Lipschitz in $x$. Prior variance reduction results (e.g. [18, 8, 25]) do not make use of this assumption. However, we we show in Appendix B that simply replacing all instances of $G$ or $G_t$ in the parameters of STORM with an oracle-tuned value of $\sigma$ allows us to dispense with this assumption while still avoiding all checkpoint gradients.

Also note that, as in similar work on stochastic minimization of non-convex functions, Theorem 1 only bounds the gradient of a randomly selected iterate [9]. However, in practical implementations we expect the last iterate to perform equally well.

Our analysis formalizes the intuition developed in the previous section through a Lyapunov potential function. Our Lyapunov function is somewhat non-standard: for smooth non-convex functions, the Lyapunov function is typically of the form $\Phi_t = F(\boldsymbol{x}_t)$, but we propose to use the function $\Phi_t = F(\boldsymbol{x}_t) + z_t \|\boldsymbol{\epsilon}_t\|^2$ for a time-varying $z_t \propto \eta_{t-1}^{-1}$, where $\boldsymbol{\epsilon}_t$ is the error in the update introduced in the previous section. The use of time-varying $z_t$ appears to be critical for us to avoid using any checkpoints: with constant $z_t$ it seems that one always needs at least one checkpoint gradient. Potential functions of this form have been used to analyze momentum algorithms in order to prove asymptotic guarantees, see, e.g., Ruszczynski and Syski [23]. However, as far as we know, this use of a potential is somewhat different than most variance reduction analyses, and so may provide avenues for further development. We now proceed to the proof of Theorem 1.

### 5.1 Proof of Theorem 1

First, we consider a generic SGD-style analysis. Most SGD analyses assume that the gradient estimates used by the algorithm are unbiased of $\nabla F(\boldsymbol{x}_t)$, but unfortunately $\boldsymbol{d}_t$ biased. As a result, we need the following slightly different analysis. For lack of space, the proof of this Lemma and the next one are in the Appendix.

**Lemma 1.** *Suppose $\eta_t \leq \frac{1}{4L}$ for all t. Then*

$$\mathbb{E}[F(\boldsymbol{x}_{t+1}) - F(\boldsymbol{x}_t)] \leq \mathbb{E}\left[-\eta_t/4\|\nabla F(\boldsymbol{x}_t)\|^2 + 3\eta_t/4\|\boldsymbol{\epsilon}_t\|^2\right] .$$

The following technical observation is key to our analysis of STORM: it provides a recurrence that enables us to bound the variance of the estimates $\boldsymbol{d}_t$.

**Lemma 2.** *With the notation in Algorithm 1, we have*

$$\mathbb{E}\left[\|\boldsymbol{\epsilon}_t\|^2/\eta_{t-1}\right] \leq \mathbb{E}\left[2c^2\eta_{t-1}^3 G_t^2 + (1-a_t)^2(1+4L^2\eta_{t-1}^2)\|\boldsymbol{\epsilon}_{t-1}\|^2/\eta_{t-1}\right.$$
$$\left. +4(1-a_t)^2 L^2\eta_{t-1}\|\nabla F(\boldsymbol{x}_{t-1})\|^2\right] .$$

Lemma 2 exhibits a somewhat involved algebraic identity, so let us try to build some intuition for what it means and how it can help us. First, multiply both sides by $\eta_{t-1}$. Technically the expectations make this a forbidden operation, but we ignore this detail for now. Next, observe that $\sum_{t=1}^T G_t^2$ is roughly $\Theta(T)$ (since the the variance prevents $\|g_t\|^2$ from going to zero even when $\|\nabla F(\boldsymbol{x}_t)\|$ does). Therefore $\eta_t$ is roughly $O(1/t^{1/3})$, and $a_t$ is roughly $O(1/t^{2/3})$. Discarding all constants, and observing that $(1 - a_t)^2 \leq (1 - a_t)$, the above Lemma is then saying that

$$\mathbb{E}[\|\boldsymbol{\epsilon}_t\|^2] \leq \mathbb{E}\left[\eta_{t-1}^4 + (1-a_t)\|\boldsymbol{\epsilon}_{t-1}\|^2 + \eta_{t-1}^2\|\nabla F(\boldsymbol{x}_{t-1})\|^2\right]$$
$$= \mathbb{E}\left[t^{-4/3} + \left(1 - t^{-2/3}\right)\|\boldsymbol{\epsilon}_{t-1}\|^2 + t^{-1/3}\|\nabla F(\boldsymbol{x}_{t-1})\|^2\right] .$$

We can use this recurrence to compute a kind of "equilibrium value" for $\mathbb{E}[\|\boldsymbol{\epsilon}_t\|^2]$: set $\mathbb{E}[\|\boldsymbol{\epsilon}_t\|^2] = \mathbb{E}[\|\boldsymbol{\epsilon}_{t-1}\|^2]$ and solve to obtain $\|\boldsymbol{\epsilon}_t\|^2$ is $O(1/t^{2/3} + \|\nabla F(\boldsymbol{x}_t)\|^2)$. This in turn suggests that, whenever $\|\nabla F(\boldsymbol{x}_t)\|^2$ is greater than $1/t^{2/3}$, the gradient estimate $\boldsymbol{d}_t = \nabla F(\boldsymbol{x}_t) + \boldsymbol{\epsilon}_t$ will be a very good approximation of $\nabla F(\boldsymbol{x}_t)$ so that gradient descent should make very fast progress. Therefore, we expect the "equilibrium value" for $\|\nabla F(\boldsymbol{x}_t)\|^2$ to be $O(1/T^{2/3})$, since this is the point at which the estimate $\boldsymbol{d}_t$ becomes dominated by the error.

We formalize this intuition using a Lyapunov function of the form $\Phi_t = F(\boldsymbol{x}_t) + z_t\|\boldsymbol{\epsilon}_t\|^2$ in the proof of Theorem 1 below.

*Proof of Theorem 1.* Consider the potential $\Phi_t = F(\boldsymbol{x}_t) + \frac{1}{32L^2\eta_{t-1}}\|\boldsymbol{\epsilon}_t\|^2$. We will upper bound $\Phi_{t+1} - \Phi_t$ for each $t$, which will allow us to bound $\Phi_T$ in terms of $\Phi_1$ by summing over $t$. First, observe that since $w \geq (4Lk)^3$, we have $\eta_t \leq \frac{1}{4L}$. Further, since $a_{t+1} = c\eta_t^2$, we have $a_{t+1} \leq \frac{ck}{4Lw^{1/3}} \leq 1$ for all $t$. Then, we first consider $\eta_t^{-1}\|\boldsymbol{\epsilon}_{t+1}\|^2 - \eta_{t-1}^{-1}\|\boldsymbol{\epsilon}_t\|^2$. Using Lemma 2, we obtain

$$\mathbb{E}\left[\eta_t^{-1}\|\boldsymbol{\epsilon}_{t+1}\|^2 - \eta_{t-1}^{-1}\|\boldsymbol{\epsilon}_t\|^2\right]$$

$$\leq \mathbb{E}\left[2c^2\eta_t^3 G_{t+1}^2 + \frac{(1-a_{t+1})^2(1+4L^2\eta_t^2)\|\boldsymbol{\epsilon}_t\|^2}{\eta_t} + 4(1-a_{t+1})^2 L^2\eta_t\|\nabla F(\boldsymbol{x}_t)\|^2 - \frac{\|\boldsymbol{\epsilon}_t\|^2}{\eta_{t-1}}\right]$$

$$\leq \mathbb{E}\left[\underbrace{2c^2\eta_t^3 G_{t+1}^2}_{A_t} + \underbrace{\left(\eta_t^{-1}(1-a_{t+1})(1+4L^2\eta_t^2) - \eta_{t-1}^{-1}\right)\|\boldsymbol{\epsilon}_t\|^2}_{B_t} + \underbrace{4L^2\eta_t\|\nabla F(\boldsymbol{x}_t)\|^2}_{C_t}\right].$$

Let us focus on the terms of this expression individually. For the first term, $A_t$, observe that $w \geq 2G^2 \geq G^2 + G_{t+1}^2$ to obtain:

$$\sum_{t=1}^{T} A_t = \sum_{t=1}^{T} 2c^2\eta_t^3 G_{t+1}^2 = \sum_{t=1}^{T} \frac{2k^3 c^2 G_{t+1}^2}{w + \sum_{i=1}^{t} G_i^2} \leq \sum_{t=1}^{T} \frac{2k^3 c^2 G_{t+1}^2}{G^2 + \sum_{i=1}^{t+1} G_i^2} \leq 2k^3 c^2 \ln\left(1 + \sum_{t=1}^{T+1} \frac{G_t^2}{G^2}\right)$$

$$\leq 2k^3 c^2 \ln(T+2),$$

where in the second to last inequality we used Lemma 4 in the Appendix.

For the second term $B_t$, we have

$$B_t \leq (\eta_t^{-1} - \eta_{t-1}^{-1} + \eta_t^{-1}(4L^2\eta_t^2 - a_{t+1}))\|\boldsymbol{\epsilon}_t\|^2 = \left(\eta_t^{-1} - \eta_{t-1}^{-1} + \eta_t(4L^2 - c)\right)\|\boldsymbol{\epsilon}_t\|^2.$$

Let us focus on $\frac{1}{\eta_t} - \frac{1}{\eta_{t-1}}$ for a minute. Using the concavity of $x^{1/3}$, we have $(x+y)^{1/3} \leq x^{1/3} + yx^{-2/3}/3$. Therefore:

$$\frac{1}{\eta_t} - \frac{1}{\eta_{t-1}} = \frac{1}{k}\left[\left(w + \sum_{i=1}^{t} G_i^2\right)^{1/3} - \left(w + \sum_{i=1}^{t-1} G_i^2\right)^{1/3}\right] \leq \frac{G_t^2}{3k(w + \sum_{i=1}^{t-1} G_i^2)^{2/3}}$$

$$\leq \frac{G_t^2}{3k(w - G^2 + \sum_{i=1}^{t} G_i^2)^{2/3}} \leq \frac{G_t^2}{3k(w/2 + \sum_{i=1}^{t} G_i^2)^{2/3}}$$

$$\leq \frac{2^{2/3} G_t^2}{3k(w + \sum_{i=1}^{t} G_i^2)^{2/3}} \leq \frac{2^{2/3} G^2}{3k^3}\eta_t^2 \leq \frac{2^{2/3} G^2}{12Lk^3}\eta_t \leq \frac{G^2}{7Lk^3}\eta_t,$$

where we have used that that $w \geq (4Lk)^3$ to have $\eta_t \leq \frac{1}{4L}$.

Further, since $c = 28L^2 + G^2/(7Lk^3)$, we have

$$\eta_t(4L^2 - c) \leq -24L^2\eta_t - G^2\eta_t/(7Lk^3).$$

Thus, we obtain $B_t \leq -24L^2\eta_t\|\boldsymbol{\epsilon}_t\|^2$. Putting all this together yields:

$$\frac{1}{32L^2}\sum_{t=1}^{T}\left(\frac{\|\boldsymbol{\epsilon}_{t+1}\|^2}{\eta_t} - \frac{\|\boldsymbol{\epsilon}_t\|^2}{\eta_{t-1}}\right) \leq \frac{k^3 c^2}{16L^2}\ln(T+2) + \sum_{t=1}^{T}\left[\frac{\eta_t}{8}\|\nabla F(\boldsymbol{x}_t)\|^2 - \frac{3\eta_t}{4}\|\boldsymbol{\epsilon}_t\|^2\right]. \quad (4)$$

Now, we are ready to analyze the potential $\Phi_t$. Since $\eta_t \leq \frac{1}{4L}$, we can use Lemma 1 to obtain

$$\mathbb{E}[\Phi_{t+1} - \Phi_t] \leq \mathbb{E}\left[-\frac{\eta_t}{4}\|\nabla F(\boldsymbol{x}_t)\|^2 + \frac{3\eta_t}{4}\|\boldsymbol{\epsilon}_t\|^2 + \frac{1}{32L^2\eta_t}\|\boldsymbol{\epsilon}_{t+1}\|^2 - \frac{1}{32L^2\eta_{t-1}}\|\boldsymbol{\epsilon}_t\|^2\right].$$

Summing over $t$ and using (4), we obtain

$$\mathbb{E}[\Phi_{T+1} - \Phi_1] \leq \sum_{t=1}^{T}\mathbb{E}\left[-\frac{\eta_t}{4}\|\nabla F(\boldsymbol{x}_t)\|^2 + \frac{3\eta_t}{4}\|\boldsymbol{\epsilon}_t\|^2 + \frac{1}{32L^2\eta_t}\|\boldsymbol{\epsilon}_{t+1}\|^2 - \frac{1}{32L^2\eta_{t-1}}\|\boldsymbol{\epsilon}_t\|^2\right]$$

$$\leq \mathbb{E}\left[\frac{k^3 c^2}{16L^2}\ln(T+2) - \sum_{t=1}^{T}\frac{\eta_t}{8}\|\nabla F(\boldsymbol{x}_t)\|^2\right].$$

Reordering the terms, we have

$$\mathbb{E}\left[\sum_{t=1}^{T}\eta_t\|\nabla F(\boldsymbol{x}_t)\|^2\right] \le \mathbb{E}\left[8(\Phi_1 - \Phi_{T+1}) + k^3 c^2/(2L^2)\ln(T+2)\right]$$

$$\le 8(F(\boldsymbol{x}_1) - F^\star) + \mathbb{E}[\|\boldsymbol{\epsilon}_1\|^2]/(4L^2\eta_0) + k^3 c^2/(2L^2)\ln(T+2)$$

$$\le 8(F(\boldsymbol{x}_1) - F^\star) + w^{1/3}\sigma^2/(4L^2 k) + k^3 c^2/(2L^2)\ln(T+2),$$

where the last inequality is given by the definition of $\boldsymbol{d}_1$ and $\eta_0$ in the algorithm.

Now, we relate $\mathbb{E}\left[\sum_{t=1}^{T}\eta_t\|\nabla F(\boldsymbol{x}_t)\|^2\right]$ to $\mathbb{E}\left[\sum_{t=1}^{T}\|\nabla F(\boldsymbol{x}_t)\|^2\right]$. First, since $\eta_t$ is decreasing,

$$\mathbb{E}\left[\sum_{t=1}^{T}\eta_t\|\nabla F(\boldsymbol{x}_t)\|^2\right] \ge \mathbb{E}\left[\eta_T\sum_{t=1}^{T}\|\nabla F(\boldsymbol{x}_t)\|^2\right].$$

Now, from Cauchy-Schwarz inequality, for any random variables $A$ and $B$ we have $\mathbb{E}[A^2]\mathbb{E}[B^2] \ge \mathbb{E}[AB]^2$. Hence, setting $A = \sqrt{\eta_T\sum_{t=1}^{T-1}\|\nabla F(\boldsymbol{x}_t)\|^2}$ and $B = \sqrt{1/\eta_T}$, we obtain

$$\mathbb{E}[1/\eta_T]\,\mathbb{E}\left[\eta_T\sum_{t=1}^{T}\|\nabla F(\boldsymbol{x}_t)\|^2\right] \ge \mathbb{E}\left[\sqrt{\sum_{t=1}^{T}\|\nabla F(\boldsymbol{x}_t)\|^2}\right]^2.$$

Therefore, if we set $M = \frac{1}{k}\left[8(F(\boldsymbol{x}_1) - F^\star) + \frac{w^{1/3}\sigma^2}{4L^2 k} + \frac{k^3 c^2}{2L^2}\ln(T+2)\right]$, to get

$$\mathbb{E}\left[\sqrt{\sum_{t=1}^{T}\|\nabla F(\boldsymbol{x}_t)\|^2}\right]^2 \le \mathbb{E}\left[\frac{8(F(\boldsymbol{x}_1) - F^\star) + \frac{w^{1/3}\sigma^2}{4L^2 k} + \frac{k^3 c^2}{2L^2}\ln(T+2)}{\eta_T}\right]$$

$$= \mathbb{E}\left[\frac{kM}{\eta_T}\right] \le \mathbb{E}\left[M\left(w + \sum_{t=1}^{T}G_t^2\right)^{1/3}\right].$$

Define $\zeta_t = \nabla f(\boldsymbol{x}_t, \xi_t) - \nabla F(\boldsymbol{x}_t)$, so that $\mathbb{E}[\|\zeta_t\|^2] \le \sigma^2$. Then, we have $G_t^2 = \|\nabla F(\boldsymbol{x}_t) + \zeta_t\|^2 \le 2\|\nabla F(\boldsymbol{x}_t)\|^2 + 2\|\zeta_t\|^2$. Plugging this in and using $(a+b)^{1/3} \le a^{1/3} + b^{1/3}$ we obtain:

$$\mathbb{E}\left[\sqrt{\sum_{t=1}^{T}\|\nabla F(\boldsymbol{x}_t)\|^2}\right]^2 \le \mathbb{E}\left[M\left(w + 2\sum_{t=1}^{T}\|\zeta_t\|^2\right)^{1/3} + M 2^{1/3}\left(\sum_{t=1}^{T}\|\nabla F(\boldsymbol{x}_t)\|^2\right)^{1/3}\right]$$

$$\le M(w + 2T\sigma^2)^{1/3} + \mathbb{E}\left[2^{1/3}M\left(\sqrt{\sum_{t=1}^{T}\|\nabla F(\boldsymbol{x}_t)\|^2}\right)^{2/3}\right]$$

$$\le M(w + 2T\sigma^2)^{1/3} + 2^{1/3}M\left(\mathbb{E}\left[\sqrt{\sum_{t=1}^{T}\|\nabla F(\boldsymbol{x}_t)\|^2}\right]\right)^{2/3},$$

where we have used the concavity of $x \mapsto x^a$ for all $a \le 1$ to move expectations inside the exponents.

Now, define $X = \sqrt{\sum_{t=1}^{T}\|\nabla F(\boldsymbol{x}_t)\|^2}$. Then the above can be rewritten as:

$$(\mathbb{E}[X])^2 \le M(w + 2T\sigma^2)^{1/3} + 2^{1/3}M(\mathbb{E}[X])^{2/3}.$$

Note that this implies that either $(\mathbb{E}[X])^2 \le 2M(w + T\sigma^2)^{1/3}$, or $(\mathbb{E}[X])^2 \le 2 \cdot 2^{1/3}M(\mathbb{E}[X])^{2/3}$. Solving for $\mathbb{E}[X]$ in these two cases, we obtain

$$\mathbb{E}[X] \le \sqrt{2M}(w + 2T\sigma^2)^{1/6} + 2M^{3/4}.$$

Finally, observe that by Cauchy-Schwarz we have $\sum_{t=1}^{T}\|\nabla F(\boldsymbol{x}_t)\|/T \le X/\sqrt{T}$ so that

$$\mathbb{E}\left[\sum_{t=1}^{T}\frac{\|\nabla F(\boldsymbol{x}_t)\|}{T}\right] \le \frac{\sqrt{2M}(w + 2T\sigma^2)^{1/6} + 2M^{3/4}}{\sqrt{T}} \le \frac{w^{1/6}\sqrt{2M} + 2M^{3/4}}{\sqrt{T}} + \frac{2\sigma^{1/3}}{T^{1/3}},$$

where we used $(a+b)^{1/3} \le a^{1/3} + b^{1/3}$ in the last inequality. $\qquad\square$

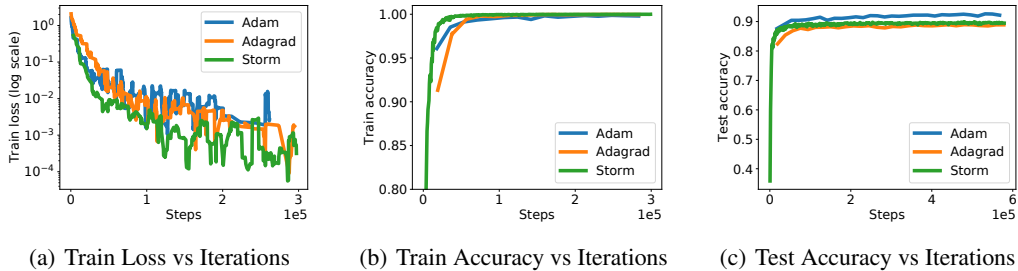

| (a) Train Loss vs Iterations | (b) Train Accuracy vs Iterations | (c) Test Accuracy vs Iterations |

Figure 1: Experiments on CIFAR-10 with ResNet-32 Network.

# 6  Empirical Validation

In order to confirm that our advances do indeed yield an algorithm that performs well and requires little tuning, we implemented STORM in TensorFlow [1] and tested its performance on the CIFAR-10 image recognition benchmark [14] using a ResNet model [10], as implemented by the Tensor2Tensor package [26][1]. We compare STORM to AdaGrad and Adam, which are both very popular and successful optimization algorithms. The learning rates for AdaGrad and Adam were swept over a logarithmically spaced grid. For STORM, we set $w = k = 0.1$ as a default[2] and swept $c$ over a logarithmically spaced grid, so that all algorithms involved only one parameter to tune. No regularization was employed. We record train loss (cross-entropy), and accuracy on both the train and test sets (see Figure 1).

These results show that, while STORM is only marginally better than AdaGrad on test accuracy, on both training loss and accuracy STORM appears to be somewhat faster in terms of number of iterations. We note that the convergence proof we provide actually only applies to the training loss (since we are making multiple passes over the dataset). We leave for the future whether appropriate regularization can trade-off STORM's better training loss performance to obtain better test performance.

# 7  Conclusion

We have introduced a new variance-reduction-based algorithm, STORM, that finds critical points in stochastic, smooth, non-convex problems. Our algorithm improves upon prior algorithms by virtue of removing the need for checkpoint gradients, and incorporating adaptive learning rates. These improvements mean that STORM is substantially easier to tune: it does not require choosing the size of the checkpoints, nor how often to compute the checkpoints (because there are no checkpoints), and by using adaptive learning rates the algorithm enjoys the same robustness to learning rate tuning as popular algorithms like AdaGrad or Adam. STORM obtains the optimal convergence guarantee, adapting to the level of noise in the problem without knowledge of this parameter. We verified that on CIFAR-10 with a ResNet architecture, STORM indeed seems to be optimizing the objective in fewer iterations than baseline algorithms.

Additionally, we point out that STORM's update formula is strikingly similar to the standard SGD with momentum heuristic employed in practice. To our knowledge, no theoretical result actually establishes an advantage of adding momentum to SGD in stochastic problems, creating an intriguing mystery. While our algorithm is not precisely the same as the SGD with momentum, we feel that it provides strong intuitive evidence that momentum is performing some kind of variance reduction. We therefore hope that some of the analysis techniques used in this paper may provide a path towards explaining the advantages of momentum.

## Footnotes

[1]`https://github.com/google-research/google-research/tree/master/storm_optimizer`

[2]We picked these defaults by tuning over a logarithmic grid on the much-simpler MNIST dataset [15]. $w$ and $k$ were not tuned on CIFAR10.

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
