[Supplementary Material]

## A  Extra Lemmas

In this section we (re)state and prove some Lemmas.

First, we provide the proof of Lemma 1, restated below for convenience.

**Lemma 1.** *Suppose $\eta_t \le \frac{1}{4L}$ for all $t$. Then*

$$\mathbb{E}[F(\boldsymbol{x}_{t+1}) - F(\boldsymbol{x}_t)] \le \mathbb{E}\left[-\eta_t/4\|\nabla F(\boldsymbol{x}_t)\|^2 + 3\eta_t/4\|\boldsymbol{\epsilon}_t\|^2\right] .$$

*Proof.* Using the smoothness of $F$ and the definition of $\boldsymbol{x}_{t+1}$ from the algorithm, we have

$$\mathbb{E}[F(\boldsymbol{x}_{t+1})] \le \mathbb{E}\left[F(\boldsymbol{x}_t) - \nabla F(\boldsymbol{x}_t) \cdot \eta_t \boldsymbol{d}_t + \frac{L\eta_t^2}{2}\|\boldsymbol{d}_t\|^2\right]$$

$$= \mathbb{E}\left[F(\boldsymbol{x}_t) - \eta_t\|\nabla F(\boldsymbol{x}_t)\|^2 - \eta_t \nabla F(\boldsymbol{x}_t) \cdot \boldsymbol{\epsilon}_t + \frac{L\eta_t^2}{2}\|\boldsymbol{d}_t\|^2\right]$$

$$\le \mathbb{E}\left[F(\boldsymbol{x}_t) - \frac{\eta_t}{2}\|\nabla F(\boldsymbol{x}_t)\|^2 + \frac{\eta_t}{2}\|\boldsymbol{\epsilon}_t\|^2 + \frac{L\eta_t^2}{2}\|\boldsymbol{d}_t\|^2\right]$$

$$\le \mathbb{E}\left[F(\boldsymbol{x}_t) - \frac{\eta_t}{2}\|\nabla F(\boldsymbol{x}_t)\|^2 + \frac{\eta_t}{2}\|\boldsymbol{\epsilon}_t\|^2 + L\eta_t^2\|\boldsymbol{\epsilon}_t\|^2 + L\eta_t^2\|\nabla F(\boldsymbol{x}_t)\|^2\right]$$

$$\le \mathbb{E}\left[F(\boldsymbol{x}_t) - \frac{\eta_t}{2}\|\nabla F(\boldsymbol{x}_t)\|^2 + \frac{3\eta_t}{4}\|\boldsymbol{\epsilon}_t\|^2 + \frac{\eta_t}{4}\|\nabla F(\boldsymbol{x}_t)\|^2\right],$$

where in the second inequality we used Young's inequality, the third one uses $\|\boldsymbol{x} + \boldsymbol{y}\|^2 \le 2\|\boldsymbol{x}\|^2 + 2\|\boldsymbol{y}\|^2$, and the last one uses $\eta_t \le 1/4L$. $\square$

This next Lemma is a technical observation that is important for the proof of Lemma 2.

**Lemma 3.**

$$\mathbb{E}\left[(\nabla f(\boldsymbol{x}_t, \xi_t) - \nabla F(\boldsymbol{x}_t)) \cdot \eta_{t-1}^{-1}(1 - a_t)^2 \boldsymbol{\epsilon}_{t-1}\right] = 0$$

$$\mathbb{E}\left[(\nabla f(\boldsymbol{x}_t, \xi_t) - \nabla f(\boldsymbol{x}_{t-1}, \xi_t) - \nabla F(\boldsymbol{x}_t) + \nabla F(\boldsymbol{x}_{t-1})) \cdot \eta_{t-1}^{-1}(1 - a_t)^2 \boldsymbol{\epsilon}_{t-1}\right] = 0 .$$

*Proof.* From inspection of the update formula, the hypothesis implies that $\boldsymbol{\epsilon}_{t-1} = \boldsymbol{d}_{t-1} - \nabla F(\boldsymbol{x}_{t-1})$ and $\boldsymbol{x}_t$ are both independent of $\xi_t$. Then, by first taking expectation with respect to $\xi_t$ and then with respect to $\xi_1, \ldots, \xi_{t-1}$, we obtain

$$\mathbb{E}\left[(\nabla f(\boldsymbol{x}_t, \xi_t) - \nabla F(\boldsymbol{x}_t)) \cdot \eta_{t-1}^{-1}(1 - a_t)^2 \boldsymbol{\epsilon}_{t-1}\right]$$
$$= \mathbb{E}\left[\mathbb{E}\left[(\nabla f(\boldsymbol{x}_t, \xi_t) - \nabla F(\boldsymbol{x}_t)) \cdot \eta_{t-1}^{-1}(1 - a_t)^2 \boldsymbol{\epsilon}_{t-1}|\xi_1, \ldots, \xi_{t-1}\right]\right] = 0 .$$

Analogously, for the second equality we have

$$\mathbb{E}\left[(\nabla f(\boldsymbol{x}_t, \xi_t) - \nabla f(\boldsymbol{x}_{t-1}, \xi_t) - \nabla F(\boldsymbol{x}_t) + \nabla F(\boldsymbol{x}_{t-1})) \cdot \eta_{t-1}^{-1}(1 - a_t)^2 \boldsymbol{\epsilon}_{t-1}\right]$$
$$= \mathbb{E}\left[\mathbb{E}\left[(\nabla f(\boldsymbol{x}_t, \xi_t) - \nabla f(\boldsymbol{x}_{t-1}, \xi_t) - (\nabla F(\boldsymbol{x}_t) - \nabla F(\boldsymbol{x}_{t-1}))) \cdot \eta_{t-1}^{-1}(1 - a_t)^2 \boldsymbol{\epsilon}_{t-1}|\xi_1, \ldots, \xi_{t-1}\right]\right]$$
$$= 0 . \qquad \square$$

The following Lemma is a standard consequence of convexity.

**Lemma 4.** *Let $a_0 > 0$ and $a_1, \ldots, a_T \ge 0$. Then*

$$\sum_{t=1}^{T} \frac{a_t}{a_0 + \sum_{i=1}^{t} a_i} \le \ln\left(1 + \frac{\sum_{i=1}^{t} a_i}{a_0}\right) .$$

*Proof.* By the concavity of the log function, we have

$$\ln\left(a_0 + \sum_{i=1}^{t} a_i\right) - \ln\left(a_0 + \sum_{i=1}^{t-1} a_i\right) \ge \frac{a_t}{a_0 + \sum_{i=1}^{t} a_i} .$$

Summing over $t = 1, \ldots, T$ both sides of the inequality, we have the stated bound. $\square$

## A.1 Proof of Lemma 2

In this section we present the deferred proof of Lemma 2, restating the result below for reference

**Lemma 2.** *With the notation in Algorithm 1, we have*

$$\mathbb{E}\left[\|\boldsymbol{\epsilon}_t\|^2/\eta_{t-1}\right] \leq \mathbb{E}\left[2c^2\eta_{t-1}^3 G_t^2 + (1-a_t)^2(1+4L^2\eta_{t-1}^2)\|\boldsymbol{\epsilon}_{t-1}\|^2/\eta_{t-1}\right.$$
$$\left. +4(1-a_t)^2 L^2\eta_{t-1}\|\nabla F(\boldsymbol{x}_{t-1})\|^2\right] .$$

*Proof.* First, observe that

$$\mathbb{E}\left[\eta_{t-1}^3\|\nabla f(\boldsymbol{x}_t,\xi_t) - \nabla F(\boldsymbol{x}_t)\|^2\right]$$
$$= \mathbb{E}\left[\eta_{t-1}^3(\|\nabla f(\boldsymbol{x}_t,\xi_t)\|^2 + \|\nabla F(\boldsymbol{x}_t)\|^2 - 2\nabla f(\boldsymbol{x}_t,\xi_t)\cdot\nabla F(\boldsymbol{x}_t))\right]$$
$$= \mathbb{E}\left[\eta_{t-1}^3\mathbb{E}\left[\|\nabla f(\boldsymbol{x}_t,\xi_t)\|^2 + \|\nabla F(\boldsymbol{x}_t)\|^2 - 2\nabla f(\boldsymbol{x}_t,\xi_t)\cdot\nabla F(\boldsymbol{x}_t)\big|\xi_1,\ldots,\xi_{t-1}\right]\right]$$
$$= \mathbb{E}\left[\eta_{t-1}^3(\|\nabla f(\boldsymbol{x}_t,\xi_t)\|^2 - \|\nabla F(\boldsymbol{x}_t)\|^2)\right]$$
$$\leq \mathbb{E}\left[\eta_{t-1}^3\|\nabla f(\boldsymbol{x}_t,\xi_t)\|^2\right] . \tag{5}$$

In the same way, we also have that

$$\mathbb{E}\left[\eta_{t-1}^{-1}(1-a_t^2)\|\nabla f(\boldsymbol{x}_t,\xi_t) - \nabla f(\boldsymbol{x}_{t-1},\xi_t) - \nabla F(\boldsymbol{x}_t) + \nabla F(\boldsymbol{x}_{t-1})\|^2\right]$$
$$\leq \mathbb{E}\left[\eta_{t-1}^{-1}(1-a_t^2)\|\nabla f(\boldsymbol{x}_t,\xi_t) - \nabla f(\boldsymbol{x}_{t-1},\xi_t))\|^2\right] . \tag{6}$$

By definition of $\boldsymbol{\epsilon}_t$ and the notation in Algorithm 1, we have $\boldsymbol{\epsilon}_t = \boldsymbol{d}_t - \nabla F(\boldsymbol{x}_t) = \nabla f(\boldsymbol{x}_t,\xi_t) + (1-a_t)(\boldsymbol{d}_{t-1} - \nabla f(\boldsymbol{x}_{t-1},\xi_t)) - \nabla F(\boldsymbol{x}_t)$. Hence, we can write

$$\mathbb{E}\left[\eta_{t-1}^{-1}\|\boldsymbol{\epsilon}_t\|^2\right] = \mathbb{E}\left[\eta_{t-1}^{-1}\|\nabla f(\boldsymbol{x}_t,\xi_t) + (1-a_t)(\boldsymbol{d}_{t-1} - \nabla f(\boldsymbol{x}_{t-1},\xi_t)) - \nabla F(\boldsymbol{x}_t)\|^2\right]$$
$$= \mathbb{E}\left[\eta_{t-1}^{-1}\|a_t(\nabla f(\boldsymbol{x}_t,\xi_t) - \nabla F(\boldsymbol{x}_t)) + (1-a_t)(\nabla f(\boldsymbol{x}_t,\xi_t) - \nabla f(\boldsymbol{x}_{t-1},\xi_t) - \nabla F(\boldsymbol{x}_t) + \nabla F(\boldsymbol{x}_{t-1}))\right.$$
$$\left. + (1-a_t)(\boldsymbol{d}_{t-1} - \nabla F(\boldsymbol{x}_{t-1}))\|^2\right]$$
$$\leq \mathbb{E}\left[2c^2\eta_{t-1}^3\|\nabla f(\boldsymbol{x}_t,\xi_t) - \nabla F(\boldsymbol{x}_t)\|^2 + 2\eta_{t-1}^{-1}(1-a_t)^2\|\nabla f(\boldsymbol{x}_t,\xi_t) - \nabla f(\boldsymbol{x}_{t-1},\xi_t) - \nabla F(\boldsymbol{x}_t) + \nabla F(\boldsymbol{x}_{t-1})\|^2\right.$$
$$\left. +\eta_{t-1}^{-1}(1-a_t)^2\|\boldsymbol{\epsilon}_{t-1}\|^2\right]$$
$$\leq \mathbb{E}\left[2c^2\eta_{t-1}^3\|\nabla f(\boldsymbol{x}_t,\xi_t)\|^2 + 2\eta_{t-1}^{-1}(1-a_t)^2\|\nabla f(\boldsymbol{x}_t,\xi_t) - \nabla f(\boldsymbol{x}_{t-1},\xi_t)\|^2 + \eta_{t-1}^{-1}(1-a_t)^2\|\boldsymbol{\epsilon}_{t-1}\|^2\right]$$
$$\leq \mathbb{E}\left[2c^2\eta_{t-1}^3 G_t^2 + 2\eta_{t-1}^{-1}(1-a_t)^2 L^2\|\boldsymbol{x}_t - \boldsymbol{x}_{t-1}\|^2 + \eta_{t-1}^{-1}(1-a_t)^2\|\boldsymbol{\epsilon}_{t-1}\|^2\right]$$
$$= \mathbb{E}\left[2c^2\eta_{t-1}^3 G_t^2 + 2(1-a_t)^2 L^2\eta_{t-1}\|\boldsymbol{d}_{t-1}\|^2 + \eta_{t-1}^{-1}(1-a_t)^2\|\boldsymbol{\epsilon}_{t-1}\|^2\right]$$
$$= \mathbb{E}\left[2c^2\eta_{t-1}^3 G_t^2 + 2(1-a_t)^2 L^2\eta_{t-1}\|\boldsymbol{\epsilon}_{t-1} + \nabla F(\boldsymbol{x}_{t-1})\|^2 + \eta_{t-1}^{-1}(1-a_t)^2\|\boldsymbol{\epsilon}_{t-1}\|^2\right]$$
$$\leq \mathbb{E}\left[2c^2\eta_{t-1}^3 G_t^2 + 4(1-a_t)^2 L^2\eta_{t-1}(\|\boldsymbol{\epsilon}_{t-1}\|^2 + \|\nabla F(\boldsymbol{x}_{t-1})\|^2) + \eta_{t-1}^{-1}(1-a_t)^2\|\boldsymbol{\epsilon}_{t-1}\|^2\right]$$
$$= \mathbb{E}\left[2c^2\eta_{t-1}^3 G_t^2 + \eta_{t-1}^{-1}(1-a_t)^2(1+4L^2\eta_{t-1}^2)\|\boldsymbol{\epsilon}_{t-1}\|^2 + 4(1-a_t)^2 L^2\eta_{t-1}\|\nabla F(\boldsymbol{x}_{t-1})\|^2\right],$$

where in the first inequality we used Lemma 3 (See Appendix A) and $\|\boldsymbol{x} + \boldsymbol{y}\|^2 \leq 2\|\boldsymbol{x}\|^2 + 2\|\boldsymbol{y}\|^2$, in the second inequality we used (5) and (6), in the third one the Lipschitzness and smoothness of the functions $f$, and in the last inequality we used again $\|\boldsymbol{x} + \boldsymbol{y}\|^2 \leq 2\|\boldsymbol{x}\|^2 + \|\boldsymbol{y}\|^2$. □

## B    Non-adaptive Bound Without Lipschitz Assumption

In our analysis of STORM in Theorem 1 we assume that the losses are $G$-Lipschitz for some known constant $G$ with probability 1. Often this kind of Lipschitz assumption is avoided in other variance-reduction analyses [18, 8, 25]. These works also require oracle knowlede of the parameter $\sigma$. It turns out that our use of this assumption is actually only necessary in order to facilitate our adaptive analysis - in fact even for ordinary (non-variance-reduced) gradient descent methods the Lipschitz assumption seems to be a common thread in adaptive analyses [16, 28]. If we are given access to the true value of $\sigma$, then we can choose a deterministic learning rate schedule in order to avoid requiring a Lipschitz bound. All that needs be done is replace all instances of $G$ or $G_t$ in STORM with the oracle-tuned value $\sigma$, which we outline in Algorithm 2 below.

The convergence guarantee of Algorithm 2 is presented in Theorem 2 below, which is nearly identical to Theorem 1 but losses adaptivity to $\sigma$ in exchange for removing the $G$-Lipschitz requirement.

**Algorithm 2** STORM without Lipschitz Bound
---
1: **Input:** Parameters $k$, $w$, $c$, initial point $\boldsymbol{x}_1$
2: Sample $\xi_1$
3: $G_1 \leftarrow \|\nabla f(\boldsymbol{x}_1, \xi_1)\|$
4: $\boldsymbol{d}_1 \leftarrow \nabla f(\boldsymbol{x}_1, \xi_1)$
5: $\eta_0 \leftarrow \frac{k}{w^{1/3}}$
6: **for** $t = 1$ **to** $T$ **do**
7:      $\eta_t \leftarrow \frac{k}{(w+\sigma^2 t)^{1/3}}$
8:      $\boldsymbol{x}_{t+1} \leftarrow \boldsymbol{x}_t - \eta_t \boldsymbol{d}_t$
9:      $a_{t+1} \leftarrow c\eta_t^2$
10:     Sample $\xi_{t+1}$
11:     $G_{t+1} \leftarrow \|\nabla f(\boldsymbol{x}_{t+1}, \xi_{t+1})\|$
12:     $\boldsymbol{d}_{t+1} \leftarrow \nabla f(\boldsymbol{x}_{t+1}, \xi_{t+1}) + (1 - a_{t+1})(\boldsymbol{d}_t - \nabla f(\boldsymbol{x}_t, \xi_{t+1}))$
13: **end for**
14: Choose $\hat{\boldsymbol{x}}$ uniformly at random from $\boldsymbol{x}_1, \ldots, \boldsymbol{x}_T$. (In practice, set $\hat{\boldsymbol{x}} = \boldsymbol{x}_T$).
15: **return** $\hat{\boldsymbol{x}}$
---

**Theorem 2.** *Under the assumptions in Section 3, for any $b > 0$, we write $k = \frac{b\sigma^{\frac{2}{3}}}{L}$. Set $c = 28L^2 + \sigma^2/(7Lk^3) = L^2(28 + 1/(7b^3))$ and $w = \max\left((4Lk)^3, 2\sigma^2, \left(\frac{ck}{4L}\right)^3\right) = \sigma^2 \max\left((4b)^3, 2, (28b + \frac{1}{7b^2})^3/64\right)$. Then, Algorithm 2 satisfies*

$$\frac{1}{T}\mathbb{E}\left[\sum_{t=1}^{T} \|\nabla F(\boldsymbol{x}_t)\|^2\right] \leq \frac{M \frac{w^{1/3}}{k}}{T} + \frac{M \frac{w\sigma^{2/3}}{k}}{T^{2/3}},$$

*where $M = 8(F(\boldsymbol{x}_1) - F^\star) + \frac{w^{1/3}\sigma^2}{4L^2 k} + \frac{k^3 c^2}{2L^2}\ln(T+2)$.*

In order to prove this Theorem, we need a non-adaptive analog of Lemma 2:

**Lemma 5.** *With the notation in Algorithm 2, we have*

$$\mathbb{E}\left[\frac{\|\boldsymbol{\epsilon}_t\|^2}{\eta_{t-1}}\right] \leq \mathbb{E}\left[2c^2\eta_{t-1}^3\sigma^2 + \frac{(1-a_t)^2(1+4L^2\eta_{t-1}^2)\|\boldsymbol{\epsilon}_{t-1}\|^2}{\eta_{t-1}}\right.$$
$$\left. +4(1-a_t)^2 L^2\eta_{t-1}\|\nabla F(\boldsymbol{x}_{t-1})\|^2\right].$$

*Proof.* The proof is nearly identical to that of Lemma 2: the only difference is that instead of using the identity $\mathbb{E}[\eta_{t-1}^3\|\nabla f(x_t, \xi_t) - \nabla F(x_t)\|^2] \leq \mathbb{E}[\eta_{t-1}^3\|\nabla f(x_t, \xi_t)\|^2] = \mathbb{E}[\eta_{t-1}^3 G_t^2]$, we directly use the value of $\sigma$: $\mathbb{E}[\eta_{t-1}^3\|\nabla f(x_t, \xi_t) - \nabla F(x_t)\|^2] \leq \eta_{t-1}^3\sigma^2$. $\square$

Now we can prove Theorem 2:

*Proof of Theorem 2.* This proof is also nearly identical to the analogous adaptive result of Theorem 1.

Again, we consider the potential $\Phi_t = F(\boldsymbol{x}_t) + \frac{1}{32L^2\eta_{t-1}}\|\boldsymbol{\epsilon}_t\|^2$ and upper bound $\Phi_{t+1} - \Phi_t$ for each $t$.

Since $w \geq (4Lk)^3$, we have $\eta_t \leq \frac{1}{4L}$. Further, since $a_{t+1} = c\eta_t^2$, we have $a_{t+1} \leq \frac{ck}{4Lw^{1/3}} \leq 1$ for all $t$. Then, we first consider $\eta_t^{-1}\|\boldsymbol{\epsilon}_{t+1}\|^2 - \eta_{t-1}^{-1}\|\boldsymbol{\epsilon}_t\|^2$. Using Lemma 5, we obtain

$$\mathbb{E}\left[\eta_t^{-1}\|\boldsymbol{\epsilon}_{t+1}\|^2 - \eta_{t-1}^{-1}\|\boldsymbol{\epsilon}_t\|^2\right]$$
$$\leq \mathbb{E}\left[2c^2\eta_t^3\sigma^2 + \frac{(1-a_{t+1})^2(1+4L^2\eta_t^2)\|\boldsymbol{\epsilon}_t\|^2}{\eta_t} + 4(1-a_{t+1})^2 L^2\eta_t\|\nabla F(\boldsymbol{x}_t)\|^2 - \frac{\|\boldsymbol{\epsilon}_t\|^2}{\eta_{t-1}}\right]$$
$$\leq \mathbb{E}\left[\underbrace{2c^2\eta_t^3\sigma^2}_{A_t} + \underbrace{\left(\eta_t^{-1}(1-a_{t+1})(1+4L^2\eta_t^2) - \eta_{t-1}^{-1}\right)\|\boldsymbol{\epsilon}_t\|^2}_{B_t} + \underbrace{4L^2\eta_t\|\nabla F(\boldsymbol{x}_t)\|^2}_{C_t}\right].$$

Let us focus on the terms of this expression individually. For the first term, $A_t$, observe that $w \geq 2\sigma^2$ to obtain:

$$\sum_{t=1}^{T} A_t = \sum_{t=1}^{T} 2c^2 \eta_t^3 \sigma^2 = \sum_{t=1}^{T} \frac{2k^3 c^2 \sigma^2}{w + t\sigma^2} \leq \sum_{t=1}^{T} \frac{2k^3 c^2}{t+1}$$
$$\leq 2k^3 c^2 \ln(T+2) .$$

For the second term $B_t$, we have

$$B_t \leq (\eta_t^{-1} - \eta_{t-1}^{-1} + \eta_t^{-1}(4L^2 \eta_t^2 - a_{t+1}))\|\boldsymbol{\epsilon}_t\|^2 = \left(\eta_t^{-1} - \eta_{t-1}^{-1} + \eta_t(4L^2 - c)\right)\|\boldsymbol{\epsilon}_t\|^2 .$$

Let us focus on $\frac{1}{\eta_t} - \frac{1}{\eta_{t-1}}$ for a minute. Using the concavity of $x^{1/3}$, we have $(x+y)^{1/3} \leq x^{1/3} + yx^{-2/3}/3$. Therefore:

$$\frac{1}{\eta_t} - \frac{1}{\eta_{t-1}} = \frac{1}{k}\left[(w + t\sigma^2)^{1/3} - (w + (t-1)\sigma^2)^{1/3}\right] \leq \frac{\sigma^2}{3k(w + (t-1)\sigma^2)^{2/3}}$$

$$\leq \frac{\sigma^2}{3k(w - \sigma^2 + t\sigma^2)^{2/3}} \leq \frac{\sigma^2}{3k(w/2 + t\sigma^2)^{2/3}}$$

$$\leq \frac{2^{2/3}\sigma^2}{3k(w + t\sigma^2)^{2/3}} \leq \frac{2^{2/3}\sigma^2}{3k^3}\eta_t^2 \leq \frac{2^{2/3}\sigma^2}{12Lk^3}\eta_t \leq \frac{\sigma^2}{7Lk^3}\eta_t,$$

where we have used that that $w \geq (4Lk)^3$ to have $\eta_t \leq \frac{1}{4L}$.

Further, since $c = 28L^2 + \sigma^2/(7Lk^3)$, we have

$$\eta_t(4L^2 - c) \leq -24L^2 \eta_t - \sigma^2 \eta_t/(7Lk^3) .$$

Thus, we obtain $B_t \leq -24L^2 \eta_t \|\boldsymbol{\epsilon}_t\|^2$. Putting all this together yields:

$$\frac{1}{32L^2} \sum_{t=1}^{T} \left(\frac{\|\boldsymbol{\epsilon}_{t+1}\|^2}{\eta_t} - \frac{\|\boldsymbol{\epsilon}_t\|^2}{\eta_{t-1}}\right) \leq \frac{k^3 c^2}{16L^2} \ln(T+2) + \sum_{t=1}^{T}\left[\frac{\eta_t}{8}\|\nabla F(\boldsymbol{x}_t)\|^2 - \frac{3\eta_t}{4}\|\boldsymbol{\epsilon}_t\|^2\right] . \quad (7)$$

Now, we analyze the potential $\Phi_t$. This analysis is completely identical to that of Theorem 1, and is only reproduced here for convenience. Since $\eta_t \leq \frac{1}{4L}$, we can use Lemma 1 to obtain

$$\mathbb{E}[\Phi_{t+1} - \Phi_t] \leq \mathbb{E}\left[-\frac{\eta_t}{4}\|\nabla F(\boldsymbol{x}_t)\|^2 + \frac{3\eta_t}{4}\|\boldsymbol{\epsilon}_t\|^2 + \frac{1}{32L^2 \eta_t}\|\boldsymbol{\epsilon}_{t+1}\|^2 - \frac{1}{32L^2 \eta_{t-1}}\|\boldsymbol{\epsilon}_t\|^2\right] .$$

Summing over $t$ and using (7), we obtain

$$\mathbb{E}[\Phi_{T+1} - \Phi_1] \leq \sum_{t=1}^{T} \mathbb{E}\left[-\frac{\eta_t}{4}\|\nabla F(\boldsymbol{x}_t)\|^2 + \frac{3\eta_t}{4}\|\boldsymbol{\epsilon}_t\|^2 + \frac{1}{32L^2 \eta_t}\|\boldsymbol{\epsilon}_{t+1}\|^2 - \frac{1}{32L^2 \eta_{t-1}}\|\boldsymbol{\epsilon}_t\|^2\right]$$

$$\leq \mathbb{E}\left[\frac{k^3 c^2}{16L^2} \ln(T+2) - \sum_{t=1}^{T} \frac{\eta_t}{8}\|\nabla F(\boldsymbol{x}_t)\|^2\right] .$$

Reordering the terms, we have

$$\mathbb{E}\left[\sum_{t=1}^{T} \eta_t \|\nabla F(\boldsymbol{x}_t)\|^2\right] \leq \mathbb{E}\left[8(\Phi_1 - \Phi_{T+1}) + \frac{k^3 c^2}{2L^2} \ln(T+2)\right]$$

$$\leq 8(F(\boldsymbol{x}_1) - F^\star) + \frac{1}{4L^2 \eta_0}\mathbb{E}[\|\boldsymbol{\epsilon}_1\|^2] + \frac{k^3 c^2}{2L^2} \ln(T+2)$$

$$\leq 8(F(\boldsymbol{x}_1) - F^\star) + \frac{w^{1/3}\sigma^2}{4L^2 k} + \frac{k^3 c^2}{2L^2} \ln(T+2),$$

where the last inequality is given by the definition of $\boldsymbol{d}_1$ and $\eta_0$ in the algorithm.

At this point the rest of the proof could proceed in an identical manner to that of Theorem 1. However, since $\eta_t$ is now idependent of $\nabla F(x_t)$ by virtue of being deterministic, we can simplify the remainder of the proof somewhat by avoiding the use of Cauchy-Schwarz inequality.

Since $\eta_t$ is deterministic, we have $\mathbb{E}\left[\sum_{t=1}^{T}\eta_t\|\nabla F(\boldsymbol{x}_t)\|^2\right] \geq \eta_T\mathbb{E}\left[\sum_{t=1}^{T}\|\nabla F(\boldsymbol{x}_t)\|^2\right]$. Then divide by $T\eta_T$ to conclude

$$\frac{1}{T}\mathbb{E}\left[\sum_{t=1}^{T}\|\nabla F(\boldsymbol{x}_t)\|^2\right] \leq \frac{M\frac{w^{1/3}}{k}}{T} + \frac{M\frac{w\sigma^{2/3}}{k}}{T^{2/3}},$$

where we have used the definition $M = 8(F(\boldsymbol{x}_1) - F^\star) + \frac{w^{1/3}\sigma^2}{4L^2 k} + \frac{k^3 c^2}{2L^2}\ln(T+2)$ and the identity $(a+b)^{1/3} \leq a^{1/3} + b^{1/3}$ ◻