[Reviews · NeurIPS 2019]

Reviewer 1



######################################################## I thank the authors for their response. Given the authors' promises to address the reviewers concerns, I continue to support acceptance of this paper. I agree with the other reviewers that a more thorough comparison with previous work such as SARAH should be added to the paper, and I think care should be taken in making the theorem statement more understandable. ######################################################## This is a very nice paper. The ideas are presented clearly and concisely. The proofs are accompanied by intuitive explanations. Most of all, the algorithm is straightforward and gives me confidence that it would be a good and robust approach to solving non-convex optimization problems in practice. The connection between momentum and variance reduction is particularly useful and may be quite useful for the analysis of other algorithms in the future. The one trouble spot is the theorem statement, which is quite confusing and difficult to parse. I found it very hard to make sense of all the parameters b, k, c, w, M, etc.

Reviewer 2



##### Thank you for your feedback. I agree with R3 that you did a poor job on relating your work to existing methods, in particular SARAH. Please also make sure that you carefully address the question of optimality. I also realized that your method in fact has nothing to do with momentum. Consider for instance deterministic objective, f(x, \xi)=f(x). If one has a tight estimate, i.e. d_{t-1}=\nabla f(x_{t-1}), then from your update rules it follows that d_t=\nabla f(x_t), i.e. the method become gradient descent with no momentum! Your title, thus, is very confusing and I highly encourage you to change it. Otherwise it will pollute the literature on real momentum methods. Please provide some basic extra experiments in the supplementary. I suggest you to 1) check the difference between uniform random sampling and random permutation 2) do some grid search to show how sensitive your method is to the parameter selection. I agree that the theory is not directly applicable here, but since you decided to do some experiments, it's better to do a proper study rather than a shallow observation that it works. ##### First of all, let me note that in work [1] there has been developed a very similar method. However, I believe it is rather a coincidence since 1) works use significantly different stepsizes, 2) work [1] uses two data samples per iteration, while this work only uses one, 3) the analysis is different. The fact that the method achieves optimal complexity for nonconvex optimization is the main reason I vote for acceptance. The method is simple, and it's clear that it brings variance reduction closer to being practical for training neural networks. The lack of experimental results disappoints me, but I still think the contribution is significant. Probably the most important part is that the authors manage to do here is working with expectations without large batches. However, based on the proof I feel that it's mostly due to assumption that all functions are almost surely Lipschitz in terms of their gradients (for instance it's used in Lemma 2, which is a variance bound, suggesting why Storm doesn't need large batches to control the variance). In contrast, SVRG and SARAH work without this assumption meaning that the results presented in this work are not strictly better. Several experimental details are missing. It's interesting what minibatch sizes the authors used, and otherwise the plot with the number of steps does not tell us how many passes over the data you made. How did you sample data during training? It's common to run SGD or Adam with random reshuffling, while your theory is for uniform sampling. Please range y-axis from 0.8 to 0.95 in Figure 1(c) to make the comparison clearer. Clearly, I wouldn't have had this questions if the authors provided their code. Are the authors going to release their code online along with values of c that they used to get the plots? Minor comment: in line 99, did you mean to write "e.g." instead of "i.e."? [1] Hybrid Stochastic Gradient Descent Algorithms for Stochastic Nonconvex Optimization

Reviewer 3



1) First, the reviewer thinks that, this algorithm combines both momentum and SARAH estimator in (Nguyen et al, 2017). So, it would be better if the authors rigorously discuss this. 2) On page 1, the authors write \nabla{F}(x) = 0 is a criterion that used in stochastic methods? I think it should be in some clear sense such as expectation, probability 1, etc. 3) I am not sure if [27] provides a lower bound complexity for expectation problems. It only considered a finite sum problem. So far I have not seen any results on lower bound complexity for the expectation problems under standard assumptions, i.e., smoothness and bounded variance. 4) The authors may be missing some recent works based on SARAH estimators such as SPIDER, SpiderBoost, and ProxSARAH which achieve optimal rate in the finite-sum case, and the best-known rate in the expectation case. It should be useful if these works are mentioned. Note that some of these methods can work with single-sample, and still achieve optimal rates. 5) I don't think it is proper to claim that STORM achieves optimal rate. First, it relies on an additional assumption: bounded gradient apart from other two assumptions. Second, I have not seen any paper studying the lower bound complexity for the expectation problems. So, it is a bit unclear if this complexity is optimal. 6) It is not clear about the assumptions in Section 3. It relies on the smoothness and bounded gradients with probability 1, which seems unclear to me to where it is used. Everything in the proof is relied on expectation. 7) The authors also claim that (page 2 and conclusion), Algorithm 1 resolves the tuning parameters, but this is not true. It also relies on several parameters, at least three: k, c, w.

[Author Response · NeurIPS 2019]

Many thanks to all the reviewers for their dedicated work and helpful comments—we will be sure to incorporate the suggestions to improve the paper!

**For Reviewer 1:** Thanks very much for your suggestions in improving the presentation of the theorem! $b$ is indeed a free parameter. It can be any positive value without harming the asymptotic rate. In fact, Theorem 1 should have said "for any $b > 0$" rather than "for some $b > 0$". In principle, one could find a value for $b$ in terms of the other problem-dependent parameters like $\sigma$ and $F(x_1) - F^\star$ that would optimize the non-dominant term of the bound, although we did not do so. The notion to simplify the presentation $M$ is a good one, we will do this in the final version.

**For Reviewer 2:** Indeed, the reference you point out is an independent work (it seems to have appeared only just before the submission deadline). It is quite interesting that the learning rates are so different!

In regards to the Lipschitz constant: We only need this assumption to support our adaptivity to $\sigma$. If instead we were given oracle knowledge of $\sigma$ (as it is often assumed in other works), then we would not need the Lipschitz assumption—notice that in Lemma 2 we do not actually use this assumption as the lemma is stated in terms of the empirically observed magnitudes of the gradients (The text of Lemma 2 does mention the Lipschitz assumption, but that is an oversight: a quick inspection shows that it was not used). We use the assumption in the algebra of the proof of Theorem 1 in order to bound expected values of sums involving terms like $\frac{\|g_t\|^2}{\eta_t^3}$. If instead we knew $\sigma$, we could set $\eta_t$ to $O((L + \sigma^2 t)^{-1/3})$ and obtain the exact same result (with a few more steps). We preferred to show a stronger and, in our opinion, more interesting result, with the additional assumption of Lipschitzness since $\sigma$ is typically unknown. However, in the final version we will add the straightforward extension to oracle tuning of the learning rates without Lipschitz assumption. Interestingly, the issue of the Lipschitzness shows up frequently in the adaptive learning literature—see for example the dual averaging version of AdaGrad [Duchi et al., 2011] or adaptive FTRL analysis [McMahan, 2017], which require known Lipschitz bounds in order to obtain adaptivity, but not for convergence.

In regards to the experiments: we used all the default parameters in the Tensor2Tensor package, including random reshuffling, a batch size of 128, and $0.0001$ weight decay constant. Zero attempt was made to tune this, so we actually suspect the default settings may mildly favor the Adam algorithm, which was the default optimizer. We concede that the theory does not perfectly apply in this problem (the training loss function of a neural net is not even smooth!), but we still think that the theoretical results provide strong motivation for practical performance.

We will add more detail on this to the text, and we *do* plan to release the code.

**For Reviewer 3:** Thanks for suggesting the references, we will happily discuss their relationship to our work! We would just briefly stress that, as far as we known, *none of these achieve adaptivity to sigma*. Also, even the methods that manage to have only one or $O(1)$ samples per iterations *still require at least one large batch in the first iteration*. Instead, both these issues are solved with our approach.

In regards to the assumption about bounded gradient with probability 1 vs in expectation: it is actually a bit tricky to go to expectation. A key place we use this assumption is bounding the term $A_t$ in the proof of Theorem 1. Here, it is used to bound $\sum_{t=1}^T G_{t+1}^2 \eta_t^{-3}$ in terms of $\sum_{t=1}^T G_{t+1}^2 \eta_{t+1}^{-3}$ with probability 1. This allows us to perform the sum without taking expectations until the end. With only an in-expectation assumption, we would need to understand $\mathbb{E}[\eta_{t+1}^{-3}]$, which is more subtle. Note that this also underscores why the Lipschitz bound is needed for adaptivity only—if we knew $\sigma$ then we would choose a deterministic schedule for $\eta$ which would then be easy to work with.

In regards to the optimality: You are right, although we match the best-known rate in the stochastic setting, our reference actually only proved optimality for the finite-sum setting. We note that their proof, in the case when the number of items in the sum is $O(\epsilon^{-2})$, involves functions that are $O(1)$ Lipschitz and so our algorithm does actually match the lower bound (in fact, this reasoning may actually provide a lower bound for the stochastic setting). We will clarify these issues in the text. Thank you for pointing it out!

As far as resolving the tuning of parameters: certainly we do not claim that STORM completely resolves this, but we do feel that our techniques significantly ameliorate the problem, since in general adaptive algorithms like AdaGrad or Adam seem to be more robust to hyperparameter selection. Also, while we state our algorithm with many parameters, as also noted by Reviewer 1, there is only one free parameter ($b$) in Theorem 1. However, we completely agree that removing all parameters is an excellent goal and we certainly hope to see such an algorithm in the future!

# References

John Duchi, Elad Hazan, and Yoram Singer. Adaptive subgradient methods for online learning and stochastic optimization. *Journal of Machine Learning Research*, 12(Jul):2121–2159, 2011.

H Brendan McMahan. A survey of algorithms and analysis for adaptive online learning. *The Journal of Machine Learning Research*, 18(1):3117–3166, 2017.


[Meta-Review · NeurIPS 2019]

The paper proposes a new momentum-based variance reduction algorithm for stochastic non-convex optimization. All reviewers are very positive about the paper. I am very happy to recommend acceptance as a poster and congratulate the authors on a nice piece of work. In the camera ready, please address the review comments, especially add the comparison with other related work as suggested by one reviewer. Regarding the lower bound result for nonconvex finite-sum optimization, the following recent paper might be of interest: Zhou, D., & Gu, Q. (2019). Lower bounds for smooth nonconvex finite-sum optimization. ICML. [This meta-review was reviewed and revised by the Program Chairs]